# EvoCF: Multi-Agent Collaboration via Agentic Memory-Driven Evolutionary Counterfactual Planning

Haotian Chi [1 5]  Zeyu Feng [2]  Xingrui Yu [2]  Linbo Luo [4]  Yew-Soon Ong [2 3]  Ivor Tsang [2 3]
Hechang Chen [1 5 †]  Yi Chang [1 5 †]  Haiyan Yin [2 †]

## Abstract

Planning collaboration strategies for multi-agent embodied systems remains a core challenge for LLM-based planners, which often fail to capture the physical and coordination constraints of real-world environments. To address this, we present **EvoCF**, an agentic memory-driven evolutionary counterfactual planning framework for discovering improved multi-agent collaboration strategies through counterfactual plan generation and evaluation. First, we propose a symbolic constraint inductor that induces reusable symbolic constraints from failures, forming an evolving rule library. Then, we propose an evolutionary counterfactual plan generator that systematically explores semantically consistent plan variants through rule-conditioned mutations, enabling robust collaboration strategies beyond short-sighted one-shot LLM plans. Finally, we design an agentic memory-grounded evaluator that ranks candidate plans using retrieval-augmented evidence, producing interpretable, constraint-aware selections. Across multi-agent embodied simulation benchmarks, EvoCF consistently discovers more robust and executable plans compared to baseline approaches. Our results demonstrate that grounding multi-agent planning in agentic memory and counterfactual reasoning significantly enhances both effectiveness and robustness.

## 1. Introduction

Enabling multiple embodied agents to collaborate effectively in complex environments is a central challenge for artificial intelligence, with far-reaching implications for robotics, household assistance, and industrial automation. Unlike single-agent planning, multi-agent collaboration requires agents not only to sequence their own actions but also to coordinate their roles, respect inter-agent dependencies, and adapt to the physical and communication constraints of real-world environments. Designing planners that can reliably discover and execute such joint strategies remains a core bottleneck in Embodied AI.

Large language models (LLMs) have recently been applied as high-level planners due to their ability to interpret natural language instructions and generalize across tasks. Early efforts have extended LLM planners to multi-agent settings through various paradigms. For example, SMART-LLM (Kannan et al., 2024) introduces a prompting framework that decomposes high-level missions into coalitions and task allocations via few-shot examples. Other work integrates LLM planning with iterative feedback loops: LLaMAR (Nayak et al., 2024a) propose an LLM-based plan–act–correct–verify loop that uses execution feedback for self-correction. Similarly, CoELA (Zhang et al., 2024) develops a modular approach that combines LLM reasoning with perception, long-term memory, and communication, enabling agents to coordinate and assist each other in a shared environment. At a larger scale, MacNet (Qian et al., 2025) organizes swarms of agents into directed acyclic graphs, revealing that performance follows a logistic growth law as team size increases. These advances demonstrate the promise of LLM-based planners for multi-agent tasks, but also highlight key limitations: current systems typically produce a single plan at a time without systematic revision, rely on heuristic corrections or centralized controllers, and rarely consider alternative task assignments or counterfactual re-planning.

These limitations are further amplified in dynamic, partially observable environments (POMDP). While in single-agent settings, incorporating feedback and re-planning improves robustness, extending such approaches (Song et al., 2023;

---

[1]School of Artificial Intelligence, Jilin University, China [2]CFAR and IHPC, Agency for Science, Technology and Research (A*STAR), Singapore [3]College of Computing and Data Science, Nanyang Technological University, Singapore [4]School of Cyber Engineering, Xidian University, China [5]Engineering Research Center of Knowledge-Driven Human-Machine Intelligence, Ministry of Education, Jilin University, China. Correspondence to: Haiyan Yin <yin_haiyan@a-star.edu.sg>, Hechang Chen <chenhc@jlu.edu.cn>, Yi Chang <yichang@jlu.edu.cn>.

*Proceedings of the 43$^{rd}$ International Conference on Machine Learning*, Seoul, South Korea. PMLR 306, 2026. Copyright 2026 by the author(s).

Guo et al., 2024; Chi et al., 2025; Liu et al., 2025; Hu et al., 2026) to multi-agent scenarios under partial observability is non-trivial. The joint action space scales exponentially with the number of agents, and effective coordination requires reasoning over uncertain observations, asynchronous actions, and long-range temporal dependencies such as task coupling and load balancing. We argue that robust multi-agent collaboration requires explicit counterfactual reasoning at planning time. In other words, agents should continuously ask "what if?": What if a different agent were assigned to this subtask? What if the order of two actions were swapped? What if a prerequisite action is skipped? Counterfactual reasoning has been explored in multi-agent reinforcement learning for credit assignment (e.g., the COMA algorithm uses a counterfactual baseline to isolate an agent's contribution (Foerster et al., 2018)). In the context of LLM-based planners (Chi et al., 2026; Liu et al., 2026), however, systematic counterfactual exploration of plan alternatives is largely absent.

In this paper, we propose EvoCF, an agentic-memory-driven evolutionary counterfactual planning framework for multi-agent embodied collaboration, where symbolic constraints induced from past experience guide counterfactual plan generation and evaluation to discover improved solutions. EvoCF grounds planning in a structured agentic memory of past executions, indexed by task context, participating agents, and observed outcomes. Given a new instance, it retrieves the most relevant traces and consolidates them into symbolic constraints that capture physical feasibility and coordination requirements. The resulting constraints are accumulated into an evolving rule library. EvoCF then performs an evolutionary counterfactual search that explores improved collaboration plans by mutating an initial plan while enforcing semantic consistency with the retrieved rules. Finally, an agentic memory-guided evaluator validates each candidate plan against retrieved experience and induced rules, and ranks candidates to select the most promising collaboration plan. To evaluate EvoCF, we use MAP-THOR(Nayak et al., 2024b), a large-scale simulation benchmark for multi-agent embodied rearrangement tasks under partial observability. It features realistic apartment layouts, diverse object configurations, and multiple agents with asynchronous actions, posing significant challenges for coordination, spatial reasoning, and task planning. By exploring diverse counterfactual variants, EvoCF discovers collaboration strategies that are more robust and adaptable than the single-shot outputs of reactive LLM planners. This work makes the following contributions:

- We introduce a symbolic constraint inductor that induces failure-grounded execution traces into reusable symbolic constraints, yielding an evolving rule library that conditions both plan generation and evaluation.

- We design an evolutionary counterfactual plan genera-

tor that systematically explores alternative joint plans by mutating action assignments and orderings. This enables the discovery of a robust collaboration plan beyond conventional one-shot LLM plans.

- We propose an agentic memory-grounded evaluator that ranks candidate plans using both learned constraints and retrieved outcomes. This evaluator leverages stored experiences to anticipate the consequences of actions and select effective multi-agent strategies.

- Through extensive experiments on multi-agent embodied simulation benchmarks, we demonstrate that EvoCF achieves 18% higher success rate over state-of-the-art baselines and significantly improves other metrics such as transport rate, coverage, balance, and planning efficiency.

## 2. Related Work

**LLM-based multi-agent planning.** Large language models have recently been extended to multi-agent planning. SMART-LLM (Kannan et al., 2024) decomposes missions into subtasks and allocations via prompting. LLaMAR (Nayak et al., 2024a) employs a plan–execute–monitor–repair cycle, boosting performance on MAP-THOR and Search&Rescue tasks. CoELA (Zhang et al., 2024) integrates LLM reasoning with perception, memory, and communication for emergent cooperation, while MacNet (Qian et al., 2025) scales collaboration through DAG-based hierarchies, showing logistic gains with larger teams. Other paradigms differ in centralization: CoNavGPT (Yu et al., 2023) produces global plans from a single GPT model, whereas RoCo (Mandi et al., 2024) assigns each agent its own LLM controller with natural-language communication.

**Memory-augmented Agents.** Memory-augmented agents improve long-horizon planning by retaining and reusing experience. Generative Agents (Park et al., 2023) and Voyager (Wang et al., 2023) demonstrate episodic recall for coherent behavior and skill acquisition. Recent work moves beyond passive recall toward *agentic memory*, where memory operations (what to store, how to organize, and when to retrieve) are integrated into the agent's behavior. For example, A-MEM (Xu et al., 2025) dynamically structures memories via indexing and linking, while EM-LLM (Fountas et al., 2025) organizes long contexts into episodic events for scalable retrieval. Most recently, AgeMem (Yu et al., 2026) operationalizes *agentic memory* by learning tool-based actions for unified long- and short-term memory management, moving beyond heuristic controllers.

**Counterfactual reasoning and plan optimization.** Recent work explores counterfactual reasoning in LLMs through diverse mechanisms for generating and evaluating alterna-

tive plans. Tree-of-Thoughts (Yao et al., 2023a) explores multiple reasoning paths, while ReAct (Yao et al., 2023b) interleaves reasoning and acting. Extensions include CFMAD, where agents debate opposing stances and a judge selects rational outcomes (Fang et al., 2025), and reflection-based methods such as Reflexion (Shinn et al., 2023) and COPPER (Bo et al., 2024), which refine decisions through self-critique or counterfactual rewards. Orthogonally, NSI (Shao et al., 2026) lifts interaction traces into logic-grounded programs, enabling agents to induce and continuously hone reusable skills with explicit control flow from past failures. These approaches highlight the value of reasoning over alternatives but remain limited to single-agent self-reflection or debate.

## 3. Methodology

### 3.1. Multi-agent Collaboration Problem

**Problem Setting**. We study embodied collaborative planning in partially observable environments, and formulate the problem as a multi-agent partially observable Markov decision process (POMDPs) $\mathcal{E} = \langle \mathcal{S}, \{\mathcal{O}_i\}_{i=1}^N, \{\mathcal{A}_i\}_{i=1}^N, \mathcal{G} \rangle$. $\mathcal{S}$ is the state space, $\mathcal{O}_i$ is the observational space of agent $i$, $\mathcal{A}_i$ is the action space of agent $i$, and $N$ is the number of agents. At timestep $t$, each agent receives a partial observation $o_t^i$, and they must coordinate with each other to achieve the shared goal $\mathcal{G}$.

Given a high-level instruction $\tau$ for task $\mathcal{G}$, the planner must synthesize suitable actions for each agent. This process unfolds as: (i) decomposing $\mathcal{G}$ into a sequence of subgoals $\{g_1, g_2, \ldots, g_K\}$, (ii) grounding each subgoal into joint actions $(a_t^1, \ldots, a_t^N)$, and (iii) coordinating execution under constraints such as temporal ordering, role specialization, and resource sharing. The core challenge is to ensure that the induced joint policy $\pi = (\pi^1, \ldots, \pi^N)$ produces strategies that are not only executable but also robust to coordination failures and environmental uncertainty.

**Modular Planning Roles.** As shown in Figure 1, our framework builds upon the modular multi-agent planning pipeline from LLaMAR (Nayak et al., 2024a), which organizes multi-agent collaboration as a sequential process of task decomposition and action allocation among agents.

**EvoCF** extends the pipeline with three instruction-guided modules that automate counterfactual planning and strengthen multi-agent collaboration: (i) the **Counterfactual Plan Generator**, which introduces evolutionary operators to explore diverse constraint-guided alternatives; (ii) the **Agentic Memory-Grounded Evaluator**, which grounds these candidates in past outcomes and symbolic constraints to assess their viability; and (iii) the **Symbolic Constraint Inductor**, which distills failure patterns into reusable rules that accumulate in agentic memory. Appendix A provides the prompt details of EvoCF.

### 3.2. Agentic Memory

In EvoCF, agentic memory drives evolutionary counterfactual planning by treating memory as an active decision mechanism rather than a passive log: it selectively stores informative executions, consolidates failures into reusable symbolic constraints, and retrieves and re-applies them at decision time to shape the counterfactual search space and rank candidate joint plans. This store–consolidate–retrieve loop provides persistent, experience-grounded guidance that improves both plan generation and evaluation over time.

**Structured Memory Representation.** We maintain a structured, cross-trial memory $\mathcal{M}$ that records outcome-grounded experience at the transition level. $\mathcal{M}$ serves as the central knowledge pool for efficient downstream retrieval, induction, and reasoning. Formally, each record is

$m = \left( \mathbf{o}_t, \mathbf{a}_t, \{\psi_t^k\}_{k=1}^M \right)$, where $\mathbf{o}_t = (o_t^1, \ldots, o_t^N)$ are agents' consecutive partially-observed states, $\mathbf{a}_t = (a_t^1, \ldots, a_t^N)$ are the joint actions, and $\{\psi_t^k\}_{k=1}^{K_t}$ is a set of *typed structural annotations* that provide multi-perspective interpretations of the experience. We use lowercase symbols for realized quantities $(\mathbf{o}_t, \mathbf{a}_t)$ and uppercase for sets/types $(\mathcal{M}, \Psi)$. Concretely, each $\psi^k \in \Psi$ may encode one of the following facets:

$$\psi^k \in \left\{ \begin{array}{l} \psi^{\text{plan}} \text{ (subgoal id, role assignment);} \\ \psi^{\text{eff}} \text{ (observed effects);} \\ \psi^{\text{pre}} \text{ (preconditions);} \\ \psi^{\text{fail}} \text{ (failure codes, induced rule fragments)} \end{array} \right\}. \quad (1)$$

Each structural element $m \in \mathcal{M}$ is annotated with calibrated metadata (e.g., confidence scores, support counts), enabling principled aggregation across experiences. This typed design transforms $\mathcal{M}$ into an *experience-grounded structured library* rather than a flat trajectory buffer, and aligns its reasoning surface with the multi-agent POMDP: preconditions and effects over $\{\mathcal{O}_i\}_{i=1}^N$, joint actions over $\{\mathcal{A}_i\}_{i=1}^N$, and role/temporal dependencies across agents.

We formalize retrieval through a compositional query operator $f_{\text{query}} : q \times \mathcal{M} \mapsto \Psi^*$, that maps a query and the memory into an aggregated set of structural facets $\Psi^* \subset \Psi$. Crucially, this operator provides a principled mechanism to retrieve, compose, and reason over counterfactual outcomes, induce symbolic constraints, and generalize coordination patterns, thereby enabling robust multi-agent planning beyond single-pass LLM outputs.

**Symbolic Constraint Induction.** We introduce a symbolic constraint induction mechanism that discovers and continually refines reusable rules from experiences and failure signals in multi-agent collaboration. The induced rules expand dynamically with new interactions, forming an evolving set that grounds counterfactual planning in structured knowledge and enables the discovery of robust collaborative strategies.

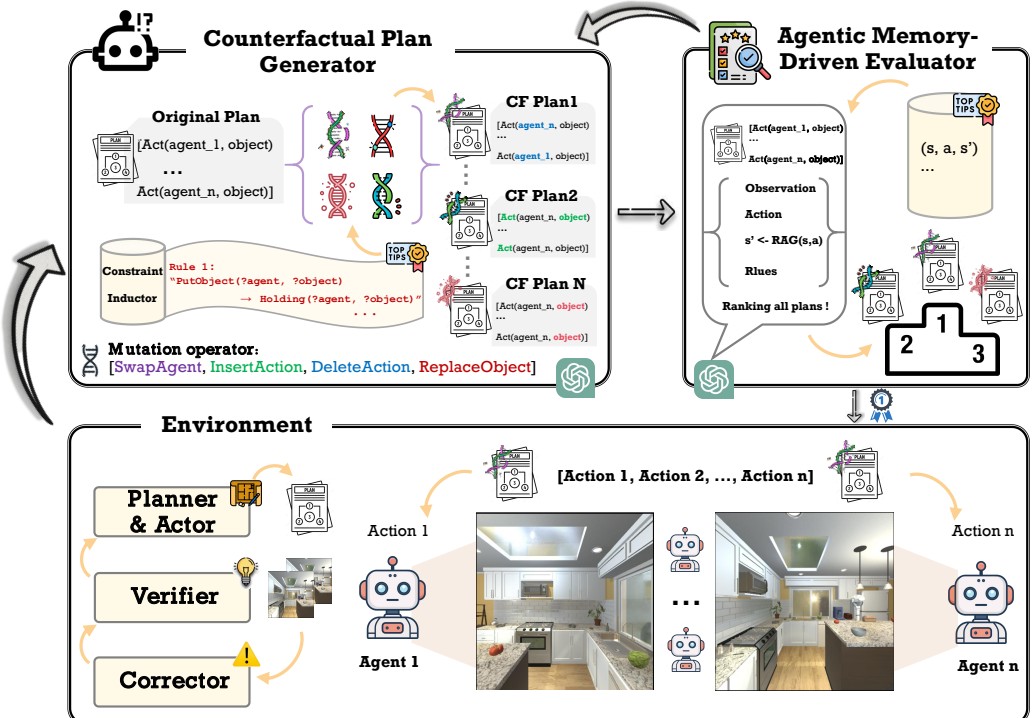

*Figure 1.* Overview of the EvoCF framework for **agentic memory-driven evolutionary counterfactual planning**. Given an initial multi-agent collaboration plan, EvoCF explores counterfactual alternatives through **evolutionary mutation operators** guided by symbolic constraints distilled from past failures. Each candidate plan is then evaluated by an **agentic memory-grounded evaluator** module that queries structured memory for relevant outcomes and constraints, to select the best one under multi-objective criteria. By integrating symbolic rule induction, evolutionary plan search, and retrieval-guided evaluation, EvoCF yields more robust and adaptive multi-agent collaboration strategies.

Formally, we define the set of symbolic constraints as $\mathcal{R}$, partitioned into two complementary categories that correspond to the dimensions of plan evaluation:

- **Task-Dependency Constraints** ($\mathcal{R}_{\text{dep}}$): capture structural preconditions and causal/temporal effects that govern task feasibility independent of collaboration. These constraints align with *single-agent execution metrics* $M_{\text{SA}}$, such as ActionValidity, ObjectReachability, and SpatialFeasibility, ensuring each agent's actions are executable and consistent with physical and semantic conditions.

- **Multi-Agent Coordination Constraints** ($\mathcal{R}_{\text{coord}}$): capture interaction-specific requirements that emerge only in collaborative settings. These constraints correspond to *multi-agent coordination metrics* $M_{\text{MA}}$, such as TemporalConsistency, LoadBalance, and GoalAlignment, ensuring effective synchronization, conflict avoidance, and cooperative goal satisfaction.

These two categories together define the rule space: $\mathcal{R} = $

$\mathcal{R}_{\text{dep}} \cup \mathcal{R}_{\text{coord}}$, providing a principled decomposition of constraints that bridges individual feasibility with collective coordination in multi-agent planning. We summarize induced constraints across six metrics in Appendix B.2.

Constraints are induced directly from memory records $m \in \mathcal{M}$ via a generator: $\mathcal{C}_{\text{gen}} : m \mapsto \{r_1, \ldots, r_k\}$ that maps each failure-annotated transition to a set of candidate rules. Each rule is annotated as $r = (\phi_r, \tau_r, \mu_r)$, where $\phi_r$ is a symbolic formula (preconditions, effects, or coordination logic), $\tau_r \in \{\text{dep}, \text{coord}\}$ is its category, and $\mu_r$ stores calibrated metadata (e.g., confidence, support). The rule set evolves online via a simple update $\mathcal{R} \leftarrow \text{Dedup}\left(\mathcal{R} \cup \mathcal{C}_{\text{gen}}(m)\right)$ ensuring compactness and continual refinement as new experiences arrive.

These rules offer a human-interpretable substrate that grounds counterfactual reasoning, significantly enhancing agents' ability to anticipate outcomes and coordinate effectively in complex multi-agent settings.

### 3.3. Evolutionary Counterfactual Planning

While one-shot LLM planning can produce plausible joint plans, it often struggles to revise early mistakes or system-

atically improve coordination under embodied feasibility constraints. EvoCF therefore frames planning as evolutionary counterfactual search: starting from an initial joint plan, it explicitly searches over counterfactual alternatives to find better collaboration strategies, with this search guided by agentic memory so that counterfactual revisions remain grounded in prior execution experience rather than unguided trial-and-error.

**Compositional Experience Retrieval.** We introduce a compositional experience retrieval framework over the agentic memory described in Sec. 3.2. A *compositional query* $q$ flexibly encodes different combinations of contextual signals, such as the current observation, a candidate action, or a subgoal tuple, optionally fused with structural identifiers (e.g., memory index or semantic tag). Concretely, we denote $q = \text{AGG}\big((x_1, x_2, \ldots, x_k), \xi\big)$, where $\text{AGG}(\cdot)$ aggregates contextual tuples and $\xi$ encodes identity information. This enables retrieval at multiple levels of granularity, ensuring that both local context and higher-order dependencies are grounded in past experience. Formally, the universal retrieval function over the structured memory is defined as:

$$R(q) = \bigcup_{m \in \mathcal{N}(q)} \Psi^{\text{rule}}(m),$$
$$\mathcal{N}(q) = \text{Top-k} \left\{ m \in \mathcal{M} \ \Big| \ \frac{\mathbf{e}(q) \cdot \mathbf{e}(m)}{\|\mathbf{e}(q)\| \, \|\mathbf{e}(m)\|} \right\}, \quad (2)$$

where $\mathbf{e}(\cdot)$ is the embedding function, $\mathcal{M}$ is episodic memory, and $\Psi^{\text{rule}}(m)$ extracts symbolic rules associated with memory $m$. This compositional formulation provides a principled mechanism to *learn from prior outcomes and induced constraints*, while offering a unified interface for both counterfactual plan generation and evaluation.

**Counterfactual Plan Generation.** EvoCF drives multi-agent collaboration beyond fixed plans by *evolving* them under memory-driven constraints. At its core, the counterfactual plan generator casts counterfactual planning as a *guided evolutionary process* that balances diversity with feasibility, ensuring candidate plans remain both task-relevant and plausibly correct. The procedure unfolds in two key steps:

(i) **Gene Representation:** A joint plan for $N$ agents at timestep $t$ is treated as an *individual*, where each agent's assigned action is modeled as a *gene*, i.e., a *receptacle* for symbolic content that can be perturbed by evolutionary operators:

$$P_t = \langle a_t^1, \ldots, a_t^N \rangle,$$
$$a_t^i \xmapsto{\text{gene}} \text{Act}(\text{agent}^i, \text{object}). \quad (3)$$

Each gene $a_t^i$ encodes an agent–action binding (e.g., `NavigateTo(Alice,Fridge)`), while the sequence $P_t$ captures the full multi-agent plan. This representation serves as a substrate for mutation operators, enabling

instruction-following exploration of counterfactual plans while respecting symbolic and memory-driven constraints.

(ii) **Mutation Operators:** EvoCF explores counterfactual diversity through a discrete set of symbolic mutation operators:

$$\Omega = \left\{ \begin{array}{l} \texttt{SwapAgent, InsertAction,} \\ \texttt{DeleteAction, ReplaceObject} \end{array} \right\}, \quad (4)$$

where each $\omega \in \Omega$ perturbs the gene sequence $P_t$ by altering agent–action bindings in a structured manner.

Unlike blind perturbations, these operators are *guided* by retrieved experience: given the current observation $\mathbf{o}_t$, the compositional retrieval in Eq. 2 returns a neighborhood of relevant memory items $\{m\}$, together with their symbolic rules $\mathcal{R}(\mathbf{o}_t)$ and outcome traces $\{\psi^{\text{out}}(m)\}$. This context steers mutations toward plausible alternatives, e.g., inserting an action known to resolve past failures, or replacing an object in line with affordances observed in similar scenes.

Formally, the counterfactual plan space is defined as:

$$\mathcal{P}_{\text{cf}}(P_t, \mathbf{o}_t) = \left\{ f_{\text{plan}}\big(P_t, \omega, \mathcal{R}(\mathbf{o}_t), \{\psi^{\text{out}}(m)\}\big) \ \Big| \ \omega \in \Omega \right\}, \quad (5)$$

where $f_{\text{plan}}$ denotes a reasoning function that conditions on retrieved constraints and outcomes to guide operator application. This design ensures that mutations remain both *diverse* and *task-relevant*, rather than arbitrary, allowing EvoCF to explore counterfactuals that better reflect feasible multi-agent collaboration strategies.

**Agentic Memory-Grounded Evaluation.** EvoCF grounds counterfactual plan evaluation in an *agentic evidence construction* process powered by compositional retrieval (Eq. 2). Rather than relying on a passive similarity lookup, it actively forms compositional queries over local state-action tuples $(\mathbf{o}_t, \mathbf{a}_t)$ to assemble a *transition-level evidence base* that is directly relevant to the current decision. This retrieval returns both realized successor outcomes $(\mathbf{o}_{t+1})$ and associated symbolic constraints $\{\psi^{\text{rule}}(m)\}$, jointly anchoring each counterfactual candidate in concrete experience while enforcing consistency with the induced rule set.

Building on this evidence, EvoCF applies an agentic memory-grounded evaluator that implements a reasoning function conditioning the fitness of each candidate on the current context, retrieved outcomes, and rules. Candidates $P'$ are ranked via:

$$\text{Rank}(P') \propto f_{\text{eval}}\big(P', \mathbf{o}_t, \mathcal{R}(\mathbf{o}_t, \mathbf{a}_t), \{\psi^{\text{out}}(m)\}\big), \ P' \in \mathcal{P}_{\text{cf}}. \quad (6)$$

Internally, $f_{\text{eval}}$ acts as a reasoning layer that integrates retrieved evidence and induced constraints to produce a constraint-aware ordering of counterfactual plans, with rule

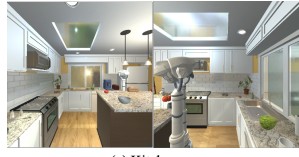
(a) Kitchen

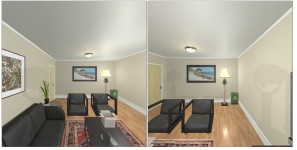
(b) LivingRoom

*Figure 2.* Photorealistic rendering of household scenarios in the MAP-THOR simulator enables the usage of multiple autonomous robots to carry out daily tasks. (a) and (b) show the egocentric views of two agents operating in the kitchen and living room.

confidences modulating the relative emphasis on feasibility, progress, and coordination. Since the ranking is invariant under any strictly monotone transformation of $f_{eval}$, only the induced ordering is relevant. This yields a *world-model-free* yet *experience-grounded* evaluator: instead of forward-simulating dynamics, EvoCF anticipates consequences by reusing outcome evidence and rule constraints from agentic memory to filter implausible strategies and prioritize promising ones. To the best of our knowledge, EvoCF is the first framework that unifies evolutionary counterfactual planning with agentic-memory-grounded evaluation, producing rankings that are simultaneously experience-grounded, symbolically constrained, and interpretable.

## 4. Experiments

To evaluate the effectiveness of our method in improving multi-agent collaboration under long-horizon and partially observable settings, we build upon the experimental framework introduced in LLaMAR (Nayak et al., 2024a). Specifically, we adopt the MAP-THOR (Nayak et al., 2024b) benchmark suite, a diverse and challenging collection of embodied household tasks based on the AI2-THOR (Kolve et al., 2017) simulator with a multi-agent setup. As shown in Figure 2, all the experiments were performed in the single-room floor plans. For consistent and fair comparison, we adopt LLaMAR's experimental setup and evaluation protocol. Concretely, we evaluate on 45 tasks in MAP-THOR, each instantiated with five room layouts spanning explicit to underspecified goals. We retain the original baselines, including Act, Chain-of-Thought (Wei et al., 2022), Re-Act (Yao et al., 2023b), SmartLLM (Kannan et al., 2024), and CoELA (Zhang et al., 2024), and report results using the same metrics defined in LLaMAR(Nayak et al., 2024a):

- **Success Rate (SR):** Fraction of episodes where all subtasks are successfully completed.
- **Transport Rate (TR):** Fraction of subtasks completed within an episode.
- **Coverage (C):** Fraction of successful interactions with target objects.
- **Balance (B):** Ratio of min/max successful high-level actions across agents, reflecting collaboration equity.

- **Average Steps (L):** Number of high-level steps taken before completion or timeout.

We use GPT-4V as the underlying VLM and compare it against the main baseline, LLaMAR, under identical simulation conditions to ensure fair and reproducible evaluation.

| Methods | SR ↑ | TR ↑ | C ↑ | B ↑ | L ↓ |
|---|---|---|---|---|---|
| Act | 0.33 | 0.67 | 0.91 | 0.59 | 24.92 |
| ReAct | 0.34 | 0.72 | 0.92 | 0.67 | 24.08 |
| CoT | 0.14 | 0.59 | 0.87 | 0.62 | 28.40 |
| SmartLLM | 0.11 | 0.23 | 0.91 | 0.45 | 29.87 |
| CoELA | 0.25 | 0.46 | 0.76 | 0.73 | 28.93 |
| LLaMAR | 0.72 | 0.93 | 0.97 | 0.85 | 20.63 |
| EvoCF (ours) | **0.84** | **0.95** | **0.99** | **0.89** | **18.69** |

*Table 1.* Comparison of evaluation metrics against baselines averaged across all tasks for the 2-agent MAP-THOR scenarios.

### 4.1. Results and Discussion

Table 1 compares our method, LLaMAR, with other baselines in a 2-agent scenario. EvoCF achieves consistent and substantial gains across all evaluation metrics, outperforming strong baselines such as LLaMAR, ReAct, and CoELA. In particular, EvoCF attains a success rate of 0.84, surpassing LLaMAR by 16%, while also improving transport rate (0.95) and coverage (0.99), indicating that agents not only complete more subtasks but also interact more reliably with task-relevant objects. These improvements are driven by EvoCF's symbolic rule induction mechanism, which extracts reusable constraints from past failures, covering both single-agent feasibility (e.g., object accessibility, action preconditions) and multi-agent coordination dependencies (e.g., spatial conflicts, role interference). These rules serve as essential structural priors that guide the generation of counterfactual plans under physical and social constraints.

| Methods | SR ↑ | TR ↑ | C ↑ | B ↑ | L ↓ |
|---|---|---|---|---|---|
| 2 | 0.84 | 0.95 | 0.99 | 0.89 | 18.69 |
| 3 | 0.87 | 0.96 | 0.99 | 0.81 | 17.36 |
| 4 | 0.82 | 0.93 | 0.99 | 0.74 | 19.51 |

*Table 2.* Benchmarking EvoCF with different numbers of agents. EvoCF exhibits stable success and coordination despite scaling complexity.

The evolutionary counterfactual planner leverages these symbolic constraints to explore alternative action assignments and subgoal orderings under the current observation, enabling the discovery of robust execution paths that preemptively avoid likely failure points. Meanwhile, the memory-driven evaluator further filters these candidates by retrieving relevant prior experiences and evaluating their long-term viability, not just based on current feasibility, but also on whether the plan aligns with patterns that have historically led to successful completions. This combination prevents overfitting to short-horizon fixes and promotes

| | SwapAgent | InsertAction | DeleteAction | ReplaceObject |
|---|---|---|---|---|
| Occurrence Frequency | 12% | 39% | 22% | 27% |
| Adoption Frequency | 8% | 17% | 8% | 10% |

*Table 3.* **Occurrence Frequency** refers to the percentage of generated counterfactual plans that include each mutation operator; **Adoption Frequency** refers to the proportion of counterfactual plans produced by each operator that are selected for execution, where 57% triggers the original plan. The results show that the counterfactual planning replaces the original plan with a noticeable high chance of 43%, highlighting the crucialness of counterfactual mutation in improving multi-agent planning robustness and flexibility.

globally coherent strategies. These capabilities contribute to the observed improvements in balance (0.89), as agents are more equitably involved in plan execution, and the reduction in average steps (18.69), as failure-prone branches are pruned early in planning. These results highlight EvoCF's capacity to integrate structured symbolic reasoning with experiential generalization for more reliable and efficient multi-agent collaboration.

To assess EvoCF's scalability, we evaluate its performance under increasing numbers of agents in the same environment. As shown in Table 2, EvoCF demonstrates improved success rate and reduced planning steps when scaling from two to four agents, indicating effective utilization of additional agent capacity through symbolic coordination. However, when scaling to four agents, performance slightly declines: the success rate drops modestly, and load balance deteriorates, reflecting increased coordination complexity. This is largely due to agent congestion and interaction overhead in shared environments, which leads to more retries and blocked plans. Additionally, the MAP-THOR benchmark enforces a fixed high-level planning step cap (L=30) to guarantee episode termination and ensure a bounded compute budget, which constrains the per-agent planning horizon as the team size grows. This may result in incomplete subtask plans, which reflects the imposed step-budget constraint rather than a limitation of our method.

| | K=1 | K=3 | K=5 | K=10 |
|---|---|---|---|---|
| SR ↑ | 0.77 | 0.81 | 0.84 | 0.82 |

*Table 4.* Sensitivity analysis on the depth of retrieved trajectories $K$ used by the evaluator.

To assess the relative impact of each mutation operator on plan quality and selection, we analyzed their usage frequency during counterfactual generation and their contribution to top-ranked plans. As shown in Table 3, these results show that InsertAction is the most impactful operator: it appears in 39% of generated plans and contributes to 17% of selected top-ranked plans. This aligns with the nature of MAP-THOR tasks, which often require fine-grained spatial adjustments (e.g., moving slightly, rotating to align) that are not captured in the initial plan. InsertAction enables such refinements by adding subtle but necessary corrections. ReplaceObject and DeleteAction provide moderate contri-

| | SR ↑ | TR ↑ | C ↑ | B ↑ | L ↓ |
|---|---|---|---|---|---|
| EvoCF (w/ Random CF Generator) | 0.68 | 0.91 | 0.97 | 0.83 | 21.37 |
| EvoCF (w/ Heuristic Evaluator) | 0.72 | 0.92 | 0.98 | 0.85 | 19.83 |
| EvoCF (w/o Constraint Inductor) | 0.75 | 0.93 | 0.98 | 0.87 | 19.62 |
| EvoCF | **0.84** | **0.95** | **0.99** | **0.89** | **18.69** |

*Table 5.* Ablation study on the impact of counterfactual plan generation, memory-driven evaluation, and Rule Induction. All metrics are evaluated on the final selected plan for each episode.

butions, useful for recovering from object choice errors or resolving collisions. SwapAgent is less frequently used, as task roles are typically well-decomposed in the initial plan.

To assess the impact of memory retrieval depth on evaluator performance, we conducted a sensitivity analysis by varying the number of retrieved trajectories $K \in \{1, 3, 5, 10\}$, while keeping all other components fixed. As shown in Table 4, increasing $K$ from 1 to 5 leads to steady improvements in success rate, as the evaluator benefits from more diverse and informative past cases. However, performance slightly drops at $K = 10$, suggesting that additional trajectories may introduce redundancy or noise. These results indicate that $K = 5$ achieves a good balance between relevance and diversity in retrieved memory, and is used as the default setting in all experiments.

We conduct an ablation study to assess the impact of three key modules in EvoCF. As shown in Table 5, using random mutations results in marginal improvement over LLaMAR, as the evaluator often falls back to the original plan when most counterfactuals are low-quality. In contrast, removing the agentic memory-driven evaluator and relying on heuristics yields better performance by selecting from structurally valid candidates, but still underperforms the full system due to a lack of experience-grounded judgment. Further, ablating the constraint inductor removes structural guidance for mutation and deprives the evaluator of reliable constraints, resulting in less targeted counterfactual plan generation and weaker plan selection. These results confirm that all three components are critical: mutation operators enable the generation of diverse and feasible counterfactuals; memory-driven evaluation ensures robust and context-aware plan selection; and rule induction extracts generalizable structural priors from past failures to guide future reasoning.

To further validate the generality and effectiveness of EvoCF, we evaluated its performance when paired with a different

| Target Task | Setting | SR ↑ | TR ↑ | C ↑ | B ↑ | L ↓ |
|---|---|---|---|---|---|---|
| Put the pots and pans on the stove | Planner Only | 0.60 | 0.88 | 0.95 | 0.80 | 20.13 |
| | w/ Transferred Constraints | 0.80 | 0.93 | 0.97 | 0.82 | 18.55 |
| Open all drawers | Planner Only | 0.40 | 0.75 | 0.94 | 0.75 | 22.41 |
| | w/ Transferred Constraints | 0.60 | 0.88 | 0.94 | 0.79 | 20.95 |

*Table 6.* Evaluating the transferability of symbolic constraints induced from the *put bread, lettuce, and tomato in the fridge* task. Constraints are injected into the planner for new tasks without using counterfactual plan generation or evaluation. Improved planning performance would indicate that symbolic constraints are reusable across tasks.

underlying language model, GPT-4o mini, and compared it against the LLaMAR baseline using the same model. We conducted experiments under the same 2-agent MAP-THOR environment and report the standard metrics.

As shown in Table 7, EvoCF consistently outperforms LLaMAR under both backbone models. With Claude 3.5 Sonnet, EvoCF improves task success, subtask completion, object interaction reliability, and collaboration balance, while requiring fewer high-level steps. This demonstrates that EvoCF can still provide clear benefits even when paired with a strong backbone model, by systematically refining initial plans through memory-guided counterfactual search. Similar improvements are observed with GPT-4o mini, where EvoCF yields even larger relative gains over LLaMAR. This suggests that EvoCF is especially beneficial for weaker planning backbones, as symbolic constraints and agentic memory-grounded evaluation help compensate for initial planning errors by generating and selecting more feasible counterfactual alternatives. Overall, these results confirm that EvoCF is model-agnostic and robust across different LLM backbones.

| | SR ↑ | TR ↑ | C ↑ | B ↑ | L ↓ |
|---|---|---|---|---|---|
| LLaMAR (Claude 3.5 Sonnet) | 0.70 | 0.92 | 0.96 | 0.84 | 21.12 |
| EvoCF (Claude 3.5 Sonnet) | 0.81 | 0.94 | 0.98 | 0.87 | 19.18 |
| LLaMAR (GPT-4o mini) | 0.61 | 0.90 | 0.96 | 0.81 | 22.34 |
| EvoCF (GPT-4o mini) | 0.74 | 0.93 | 0.98 | 0.85 | 20.11 |

*Table 7.* Backbone generalization results on 2-agent MAP-THOR scenarios. EvoCF consistently improves over LLaMAR when paired with both Claude 3.5 Sonnet and GPT-4o mini.

These results highlight that the performance improvements are brought by EvoCF itself: introducing Agentic Memory-Driven Evolutionary Counterfactual Planning, EvoCF consistently produces more feasible, coordinated, and efficient multi-agent plans across different LLM backbones. Additionally, we report the per-module latency and token cost of EvoCF in Appendix B.1.

## 4.2. Case Studies

To isolate the reusability of symbolic constraints across tasks, we design a minimal intervention experiment: we induce constraints from a single source task (`put bread, lettuce, and a tomato in the fridge`), and

inject them as structured planning priors when solving unrelated tasks (`put the pots and pans on the stove burners, open all drawers`).

These constraints are provided to the planner without invoking the counterfactual generator or evaluator, and no new rules are induced for the test tasks. Thus, any improvement in task completion indicates that the original constraints encode transferable physical and coordination knowledge applicable across tasks.

As shown in Table 6, injecting transferred constraints consistently improves task performance across all key metrics. For instance, the success rate of putting the pots and pans on the stove increases from 0.60 to 0.80, and on open all drawers from 0.40 to 0.60. Transport rate, collaboration balance, and average steps also improve, with minimal or no degradation in coverage. These improvements arise despite the constraints being induced from a structurally different task, supporting the hypothesis that EvoCF induces structurally generalizable rules.

This case study confirms that EvoCF's symbolic constraints encode transferable knowledge about action feasibility and multi-agent coordination. Even when applied to unseen tasks with different object semantics and room layouts, the rules provide effective priors for generating more executable and balanced joint plans, highlighting their potential for reuse in continual and multi-task embodied planning.

Additionally, we present a case study on the **Turn on all stove knobs** task in MAP-THOR, where agents Alice and Bob collaboratively explore and operate stove knobs throughout the kitchen scenario. As shown in Figure 3, we illustrate how EvoCF improves multi-agent collaboration via agentic memory-driven counterfactual planning.

**Symbolic Constraints Induction.** We demonstrate how EvoCF improves multi-agent planning by analyzing Step 10 with subtask *Open All Drawers*. The plans at Step 9 are: `[NavigateTo(Stove_2), OpenObject(Stove_1)]`. Execution feedback shows that Alice failed to reach `Stove_2` due to obstruction, while Bob's open attempt failed because he was not properly aligned with `Stove_2`. EvoCF captures this transition in memory and annotates it with structured outcomes. From

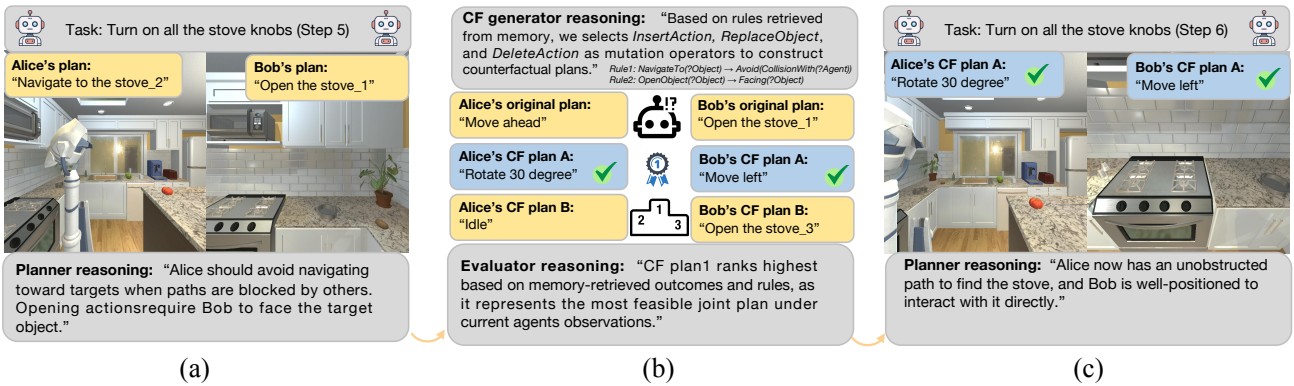

*Figure 3.* Illustration of EvoCF's reasoning process on the *Turn on all stove knobs* task. (a) The original plan at Step 5 fails: Alice attempts to navigate toward *stove_2* but is blocked, and Bob fails to open *stove_1* due to misalignment. (b) Guided by the memory-retrieved rules, EvoCF generates multiple counterfactual plans and evaluates them by an evaluator. *CF Plan A* is selected as the top-ranked plan to interact with the environment. (c) After executing *CF Plan A*, the agents are in favorable states: Alice has an unobstructed path for further exploration, and Bob is well-positioned to interact with the stove knob, enabling successful task completion.

the failure signals, it induces the following reusable symbolic constraints:

- NavigateTo(?Object) → Avoid(CollisionWith(?Agent))
- OpenObject(?Object) → Facing(?Object)

These constraints reflect spatial feasibility and interaction prerequisites that were violated in this step. Once stored in the evolving rule library, they serve as useful prior to guide future plans.

**Counterfactual Plan Generation.** The planner generates an **Original Plan** ([Move(ahead), OpenObject(Stove_1)]) in Step 6 that avoids the prior errors. The counterfactual plan generator mutates the original plan under the above constraints to generate alternative joint plans. For Step 6, it proposes the following counterfactual variants:

- **CF Plan A:** [Rotate(30), Move(left)] — Alice rotates to another angle to reduce collision risk, Bob moves left to get closer to and face the target.
- **CF Plan B:** [Idle, Move(left)] — Alice pauses, Bob retries the open action with another object.

Each counterfactual plan applies specific mutation operators based on the retried symbolic constraints: CF Plan A applies InsertAction to both Alice's and Bob's actions. CF Plan B applies DeleteAction to Alice's action and InsertAction to Bob's action.

**Agentic Memory-Grounded Evaluation.** Rather than scoring plans heuristically, EvoCF retrieves past transitions with similar object locations and interaction failures. For example, prior failures of OpenObject due to misalignment support the constraint Facing(Object), while collision cases inform the need for repositioning before navigation.

EvoCF then ranks candidate plans by combining symbolic constraints with experience-grounded outcomes. At Step 6, **CF Plan A**, [Rotate(30), Move(left)], is selected as most promising. This plan outperforms the original and other variants by jointly resolving spatial interference and misalignment: Alice takes a clearer path more aligned with the task goal, while Bob adjusts orientation, improving execution success and subgoal coordination. This case illustrates how EvoCF derives effective, failure-aware collaboration strategies through its integrated planning pipeline.

## 5. Conclusion

In this paper, we introduce a framework for discovering improved multi-agent collaboration plans via agentic memory-driven evolutionary counterfactual planning. Counterfactual planning turns plan improvement into an explicit "what-if" search problem, enabling the discovery of alternative candidates to replace an initial plan that may contain coordination mistakes or violate embodied feasibility constraints. Agentic memory guides entire planning by inducing failure experiences into symbolic constraints, maintaining an evolving rule library, and retrieving relevant rules compositionally to guide both counterfactual plan generation and evaluation. Empirical results across multi-agent collaboration benchmarks show significant gains in success rate, coordination, and efficiency.

Our future work will extend counterfactual reasoning from action assignments to subtask-level planning, enabling finer-grained edits such as goal reordering, subgoal decomposition, and partial plan substitution. We also aim to develop more expressive evaluation mechanisms, including outcome verification and deliberative agents capable of self-reflection and multi-step plan refinement, to further improve the reliability and adaptability of collaborative planning.

## Acknowledgments

This research is supported by the National Research Foundation, Singapore under its AI Singapore Programme (AISG Award No: AISG-NMLP-2024-003), and the National Research Foundation, Singapore and Infocomm Media Development Authority under its Trust Tech Funding Initiative, and in part by the New Cornerstone Science Foundation through the XPLORER PRIZE; the National Natural Science Foundation of China (No. U2341229, No. 62476110); the National Key R&D Program of China (No. 2023YFF0905400, No. 2021ZD0112500); the Key R&D Project of Jilin Province (No. 20240304200SF); the Key Research and Development Program of Shaanxi Province (No. 2025GH-YBXM-020). Any opinions, findings and conclusions or recommendations expressed in this material are those of the author(s) and do not reflect the views of the National Research Foundation, Singapore, the Agency for Science, Technology and Research, or the Infocomm Media Development Authority.

## Impact Statement

EvoCF advances multi-agent embodied planning with potential applications in household robotics, assistive technology, and industrial automation. Its interpretable symbolic constraints support human oversight, as operators can inspect and audit the rules governing agent behavior, which is a meaningful safety property for real-world deployment.

Broader adoption of capable multi-agent systems nonetheless warrants careful consideration. Robustness guarantees demonstrated on simulation benchmarks may not transfer reliably to diverse real-world environments, and systems operating in shared physical spaces introduce safety risks if coordination failures occur in proximity to humans. We encourage future work to address safety certification and human-in-the-loop oversight prior to deployment in sensitive contexts.

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

# Appendix

## A. Prompting Details

We describe the prompts used for each of the modules used in EvoCF:

---

**Symbolic Constraint Inductor**

```
<inputs> 1.5em0em {subtask}
{observation}
{action}
{failure_reason} </inputs>
```

You are a symbolic reasoning agent helping a team of {num_agents} embodied agents improve their multi-agent collaboration and planning by learning from individual failures.

Your task is to analyze one failure experience and induce symbolic constraints that, if satisfied, would have prevented the failure. These symbolic constraints help guide future planning and execution, and should be categorized into one of the following six types:

# Constraint Type Options:
1. Multi-Agent Coordination Metrics:
**Temporal Consistency**: Ensure agents follow proper semantic action order.
**Load Balance**: Avoid redundancy or idleness across agents.
**Goal Alignment**: Ensure agents' goals are mutually compatible.

2. Single-Agent Execution Metrics:
**Action Validity**: Action must align with the agent's current state.
**Object Reachability**: The object/location must be spatially reachable.
**Spatial Feasibility**: The action must be physically possible, not blocked.

## Output must follow this format:
{
symbolic_rule: `<a logic-style symbolic expression capturing the structural requirement.>`,
description: `< a plain natural language explanation of the rule.>`,
type: `<the option of constraint>`
}

---

**Counterfactual Plan Generator**

```
<inputs> 1.5em0em {subtask}
{observation}
{original_plan}
{constraints} </inputs>
```

You are a counterfactual plan generator for a team of {num_agents} embodied multi-agent robots collaboration.

Your goal is to propose alternative multi-agent joint action plans based on the current timestep's context. These plans are meant to explore meaningful variations over the original plan, guided by symbolic constraints and past failure experiences.

You will be given:
**Subtasks**: The current goal or sub-goals assigned to each agent.
**Original plan**: The current joint plan consists of one action per agent.
**Observations**: The latest environmental observations available to each agent.
**Constraints**: A set of symbolic constraint rules retrieved from past failure cases.

You are only allowed to select an action from the following predefined robot action space:

["navigate to object $< object\_id >$", "rotate in $< rotation >$ direction", "pick up object $< object\_id >$", "put object on $< receptacle_id >$", "open object $< object\_id >$", "close object $< object\_id >$", "slice object $< object\_id >$", "toggle object $< object\_id >$ on", "toggle object $< object\_id >$ off", "clean object $< object\_id >$", "look up by angle $< angle >$", "look down by angle $< angle >$", "move in $< translation >$ direction", "stay idle"]

Here, "stay idle" is used when you want the robot to stay idle for one time step. This could be used to wait for the other robot to complete its subtask. Use it only when you think it is necessary. Here $< rotation >$ can be one of ["Right", "Left"]. Here $< angle >$ is the angle in degrees and can only be one of [30, 60, 90, 120, 150, 180]. Here $< translation >$ can be one of ["Ahead", "Back", "Left", "Right"].

You must generate **N counterfactual joint action plans**, where each plan is a variation of the current original plan 'A' created by applying **one or more mutation operators** to one or more agents. The mutation operators available to you include:

- **SwapAgent**: Swap the actions assigned to agents.
- **InsertAction**: Replace the action verb, keeping the object the same.
- **DeleteAction**: Set an agent to do nothing at this timestep (stay idle).
- **ReplaceObjective**: Replace the object argument of the action, keeping the verb the same.

You must:

---

- Use retrieved constraint rules and agent's current observations to infer which agent actions are likely to lead to constraint violations or failure, based on similar past failure experiences.
- For the action(s) identified above, select appropriate mutation operators to explore counterfactual alternatives that could avoid these failures or improve success likelihood. - Ensure that each generated plan remains **semantically consistent** with the subtask.
- Ensure that each action is from the available action space listed above.
- Avoid generating plans that violate known constraints unless explicitly exploring failure.

## Format your output as a list of candidate plans:
{
1. [< Action >, < Action >, ..., < Action >],
...
N. [< Action >, < Action >, ..., < Action >]
}

## Agentic Memory-Grounded Evaluator

```
<inputs> 1.5em0em {subtask}
{observation}
{candidate_plans}
{retrieved_constraints}
{retrieved_outcomes} </inputs>
```

You are a collaborative reasoning assistant helping a team of {num agents} embodied robots evaluate possible collaboration strategies at the current time step.

You are given:
**Subtasks**: The current high-level subtask.
**Observations**: Each agent's local observation.
**Candidate plans**: A list of candidate joint plans, each consisting of one action per agent.
**Constraints**: A set of symbolic rules retrieved from memory.
**Outcomes**: A set of outcomes that provide multi-perspective interpretations of the experiences from similar situations.

Your goal is to **reason over the retrieved symbolic rules and outcomes**, and **select the most promising plan** for the current context. **You must not assign numerical scores or simulate the environment**. Instead, base your decision on:
- Which plan **best satisfies the retrieved symbolic constraints**?
- Which plan **avoids failure patterns seen in retrieved outcomes**?
- Which plan is **most likely to lead to progress** based on past experiences?

You should:
- Use multi-step reasoning to analyze how each plan aligns (or conflicts) with the constraints and past outcomes.
- Consider both **single-agent feasibility** and **multi-agent coordination**.
- Highlight which constraints each plan violates or satisfies.
- Rank the plans purely by reasoning and justify your choice.

## Output must follow this format:
{
"plan_ranking": [
{
"plan": [< Action >, ..., < Action >],
"reasoning": " < explanation > "
},
...]
}

# B. Additional Experiment Results

## B.1. Computation Overhead Analysis

As shown in table 8 and table 9, we report the per-module latency and token cost of EvoCF compared to the LLaMAR baseline.

| # Latency | LLaMAR-Moudles | Inductor | CF Generator | Evaluator | Total |
|---|---|---|---|---|---|
| LLaMAR | 7.61 | - | - | - | 166.43 |
| EvoCF (ours) | 7.61 | 1.46 | 2.17 | 2.31 | 269.76 |

*Table 8.* Average per-step latency (seconds) by module and total runtime per episode.

While EvoCF introduces additional modules beyond LLaMAR, it reduces planning steps per episode ($21.87 \rightarrow 18.69$), helping amortize the cost across fewer, higher-quality decisions. In other words, while EvoCF introduces slightly higher per-step latency and token cost, it enables faster and more reliable task completion overall, due to fewer total planning rounds and higher plan success.

| # Token | LLaMAR-Moudles | Inductor | CF Generator | Evaluator | Total |
|---|---|---|---|---|---|
| LLaMAR | 1.58K | - | - | - | 35.65K |
| EvoCF (ours) | 1.58K | 2.45K | 4.27K | 5.12K | 47.82K |

*Table 9.* Average per-step token usage by module and cumulative total per episode.

### B.2. Table of Symbolic Constraints Induced

Table 10 summarizes the symbolic constraints that were automatically induced based on failure feedback across different collaboration tasks. These results comprehensively cover the six key metrics introduced in Section 3.2, which fall under the two major categories of *Task-Dependency Constraints* and *Multi-Agent Coordination Constraints*.

| Task | Symbolic Constraints |
|---|---|
| ActionValidity | `BlockedPath`(?agent, ?destination) → ExploreOrReport(?agent); 
 ¬`Nearby`(?agent, ?object) → ChangeViewAngle(?agent, ?angle); 
 `CheckLightStatus`(?agent) → CorrectCommand(?agent); 
 `Clean`(?agent, ?object) → Holding(?agent, ?object); |
| ObjectReachability | `AccessObject`(?agent, ?object) → ClearPath(?agent, ?object); 
 `AccessObject`(?agent, ?object) → Open(?container); 
 `CleanObject`(?agent, ?object) → AtLocation(?agent, ?object); 
 `CleanObject`(?agent, ?object) → CloseTo(?agent, ?object); |
| SpatialFeasibility | `Access`(?agent, ?location) → ¬Blocked(?agent, ?location); 
 `AdjustPosition`(?agent) → ClearPath(?agent, ?destination); 
 `AdjustPosition`(?agent, ?destination) → Facing(?agent, ?destination); 
 `Align`(?agent, ?object) → ClearPath(?agent, ?object); 
 `Aligned`(?agent, ?object, ?target) ∧ CloseEnough(?agent, ?target) → Interact(?agent, ?object); 
 `Aligned`(?agent, ?target) → Move(?agent, ?direction); |
| TemporalConsistency | `Explore`(?agent) → Rotate(?agent) ∧ Move(?agent); 
 `Explore`(?agent, ?area) → Open(?agent, ?drawer); 
 `Interacting`(?agent, ?object) ∧ Interacting(?agent, ?object) → Interference(?agent, ?agent); 
 `Open`(?agent, ?object) → Before(PutObject(?agent, ?object)); 
 `OpenObject`(?agent, ?drawer) → Occupied(?drawer); |
| LoadBalance | `Assist`(?agent1, ?agent2) → Idle(?agent1) ∧ NeedAssistance(?agent2); 
 `BetterPosition`(?agent1, ?agent2, ?object) → Assist(?agent1, ?agent2); 
 `Closer`(?agent1, ?object) ∧ CanOpen(?agent1, ?object) → Open(?agent1, ?object); 
 `Closer`(?agent1, ?location) ∧ Blocked(?agent2, ?location) → Help(?agent1, ?agent2); |
| GoalAlignment | `Assist`(?agent1, ?agent2) → ClearSpace(?agent1, ?agent2); 
 `Assist`(?agent1, ?agent2, ?task) → GoalAlignment(?agent1, ?task); 
 `Assist`(?agent1, ?agent2, ?task) → Near(?agent1, ?task); 
 `Assist`(?agent1, ?agent2, Locate(?object)) → Near(?agent1, ?object); |

*Table 10.* Induced symbolic constraints across MAP-THOR tasks.

