# OpenReview forum: "EvoCF: Multi-Agent Collaboration via Agentic Memory-Driven Evolutionary Counterfactual Planning"
_ICML.cc/2026/Conference — ICML 2026 regular_

### Official Review · Reviewer_K9u1 · 2026-03-12

**Soundness:** 3
**Presentation:** 3
**Significance:** 4
**Originality:** 3
**Overall Recommendation:** 4
**Confidence:** 4

**Summary:**

This paper presents EvoCF, an evolutionary counterfactual planning framework designed to enhance multi-agent collaboration in embodied AI environments. The authors address a critical failure in current LLM-based planners: the inability to account for fine-grained physical and coordination constraints (e.g., spatial interference or temporal dependencies). The system operates through two core components: a Symbolic Constraint Inductor, which extracts reusable logic rules from failed attempts, and an Evolutionary Counterfactual Plan Generator, which explores "what-if" scenarios to find optimal paths. By maintaining an evolving memory-driven rule library, EvoCF allows agents to "learn" from past coordination errors, significantly improving task success rates in complex simulators like AI2-THOR and VirtualHome.

**Compliance With Llm Reviewing Policy:**

Affirmed.

**Key Questions For Authors:**

none

**Limitations:**

The authors explicitly mention the limitations regarding the reliance on the environment's symbolic feedback for constraint induction and the increased token consumption/latency associated with generating multiple counterfactual plans for each failure point.

**Strengths And Weaknesses:**

The work is technically sound, as it provides a rigorous closed-loop system where failure leads to symbolic induction, which in turn constrains future search spaces, backed by consistent performance gains across multiple embodied benchmarks. Its presentation is excellent; the authors provide a clear formalization of the counterfactual planning process and use intuitive examples (such as robot path conflicts) to ground the theoretical components. In terms of significance, the paper addresses the highly relevant problem of "hallucinated planning" in physical AI, offering a practical bridge between high-level LLM reasoning and low-level physical execution. The originality is high, particularly in its neuro-symbolic approach that treats coordination constraints as an evolvable memory, moving beyond simple one-shot prompting or standard reinforcement learning by utilizing the "counterfactual" reasoning capabilities of LLMs to prune inefficient collaboration strategies. However, a potential weakness lies in the computational latency introduced by the iterative counterfactual generation, which may limit the framework's applicability to high-speed, real-time robotics without further optimization of the LLM inference pipeline.

---

> ### Author Rebuttal · Authors · 2026-03-31
>
> We thank the reviewer for the positive assessment and for recognizing the value of our neuro-symbolic approach to mitigating hallucinated planning in embodied AI. We also appreciate the acknowledgment of the framework’s technical soundness and the clarity of our counterfactual planning formulation.
>
> ---
> **Weakness 1:**
> >*A potential **weakness lies in the computational latency** introduced by the iterative counterfactual generation...*
>
> **Response**:
>
> We agree that EvoCF introduces additional per-step inference overhead. However, this overhead is **largely amortized by a reduced decision horizon**, as improved planning quality leads to fewer replanning cycles and shorter execution trajectories.
>
> To validate this, we conduct additional experiments **comparing EvoCF with LLaMAR under GPT-4o and GPT-4o mini**. As shown in Table 1, EvoCF with GPT-4o-mini outperforms  LLaMAR  with GPT-4o, while also requiring fewer planning steps.
>
> |Table 1|SR|TR|C|B|L|
> |-|-|-|-|-|-|
> |LLaMAR (GPT-4o)|0.72|0.92|0.98|0.83|20.85|
> |LLaMAR (GPT-4o mini)|0.61|0.90|0.96|0.81|22.34|
> |EvoCF (GPT-4o mini)|0.74|0.93|0.98|0.85|20.11|
>
> We further measure end-to-end latency:
> |Table2 (latency)|LLaMAR-Moudles|Constraint Inductor|Counterfactual Plan Generator|Evaluator|Step|Toal|
> |-|-|-|-|-|-|-|
> |LLaMAR(GPT-4o)|7.61|-|-|-|20.85|158.66|
> |EvoCF(GPT-4o mini)|5.13|0.67|1.08|1.36|20.11|165.71|
>
> The latency analysis shows that although EvoCF introduces **additional reasoning modules, the total execution time increases by only 4.4%**, due to the reduced number of planning steps.
>
> Overall, these results suggest that the latency overhead of EvoCF is **modest and manageable** in the high-level planning setting we consider, rather than a prohibitive limitation.
>
> ---
> **Limiation 1:**
> >*The authors explicitly mention the limitations regarding the **reliance on the environment's symbolic feedback for constraint induction.***
>
> **Response**:
>
> We thank the reviewer for this insightful comment. EvoCF does not rely on idealized symbolic feedback. Instead, it **transforms raw interaction signals** such as failed action outcomes and task incompletion **into structured coordination knowledge**. This enables the system to formalize lessons learned from experience, a capability that is largely absent in existing multi-agent planning frameworks, which typically treat memory as a passive event log rather than an active decision substrate.
>
> Concretely, the Symbolic Constraint Inductor performs **LLM-based abstraction over interaction experience**, rather than simple rule extraction. Using a **lightweight  schema** with **few-shot prompting**, it automatically **infers reusable symbolic constraints directly from failure signals**, such as violated preconditions or coordination conflicts, without requiring comprehensive predefined rules. As a result, the induced constraints enable EvoCF to capture **coordination patterns** generalizable across tasks and environments.
>
> We acknowledge that induced constraint quality depends on signal fidelity, and robustness under noisy or partial feedback remains future work.
>
> ---
>
> **Limiation 2:**
> >*The **increased token consumption/latency** associated with generating multiple counterfactual plans for **each failure point.***
>
> **Response**:
>
> We thank the Reviewer for highlighting this practical trade-off. While EvoCF introduces additional reasoning steps, this compute is **strategically targeted rather than brute-force**. Counterfactual candidates are generated by applying structured mutations to the specific actions identified as failure-prone, **grounded in retrieved symbolic constraints and current observations**, rather than through redundant or unconstrained plan sampling.
>
> **`Figure 3（Line 385）`** shows this qualitatively, and **`Table 3（Line 330）`** shows these counterfactuals are materially used rather than collapsing to duplicates, with non-trivial adoption across all four mutation families and a **43% replacement rate** over the original plan. This confirms that the added compute is not merely overhead, but a worthwhile investment that converts extra reasoning into more effective and efficient planning.
>
> Furthermore, as detailed in our **`Response to Weakness 1`**, we have included new experiments using a smaller backbone (**GPT-4o-mini**). These results demonstrate that **EvoCF achieves comparable or superior performance to the LLaMAR baseline (on GPT-4o) with only a 4.4% increase in total end-to-end latency**. This proves that EvoCF’s gains stem from superior reasoning architecture rather than brute-force computation, making the framework both effective and computationally viable.
>
> ---
> We believe our results confirm that EvoCF offers a robust trade-off between reasoning quality and execution efficiency, and we look forward to addressing any remaining technical questions that may assist in your final assessment of our work.

---

> > ### Author Rebuttal · Reviewer_K9u1 · 2026-04-04
> >
> > Thank you for your reply. I have no further questions.

---

> > > ### Author Response · Authors · 2026-04-04
> > >
> > > We sincerely thank you for your valuable review. Your comments have helped us better clarify several important aspects of the paper. We're glad the revisions addressed your concerns, and appreciate your support.

---

### Official Review · Reviewer_hm4U · 2026-03-12

**Soundness:** 3
**Presentation:** 3
**Significance:** 2
**Originality:** 3
**Overall Recommendation:** 4
**Confidence:** 4

**Summary:**

This paper proposes EvoCF, a framework for multi-agent embodied collaboration planning that integrates three modules: a Symbolic Constraint Inductor (learning reusable rules from failures), an Evolutionary Counterfactual Plan Generator (exploring plan variants via mutation operators), and an Agentic Memory-Grounded Evaluator (ranking candidates via retrieval-augmented evidence). The paper evaluates on MAP-THOR (45 tasks, 2–4 agents), claiming 18% SR improvement over LLaMAR and consistent gains across transport rate, coverage, balance, and planning efficiency.

**Compliance With Llm Reviewing Policy:**

Affirmed.

**Final Justification:**

The authors' rebuttal satisfactorily addressed my concerns, leading me to revise my assessment favorably.

**Key Questions For Authors:**

1. Were baseline results re-run with GPT-4o? If not, what is LLaMAR's SR under GPT-4o alone, without EvoCF's modules?
2. How does the system behave when the optimal strategy requires a coordination pattern outside the four predefined mutation operators? Is there any evidence of strategies discovered beyond these pre-defined bounds?
3. Have you evaluated on any benchmark beyond MAP-THOR—even preliminary results on a second domain would substantially strengthen the generalization claim?

**Limitations:**

1. All experiments confined to MAP-THOR single-room household tasks; no cross-domain evidence
2. Uncontrolled GPT-4V vs. GPT-4o model discrepancy in baseline comparison
3. Mutation operators and constraint types are entirely manually designed with no learning component
4. No failure case analysis or systematic error categorization
5. Scalability beyond 4 agents remains unvalidated

**Strengths And Weaknesses:**

Main Strengths
1. The overall system design is coherent and well-motivated: grounding counterfactual multi-agent planning in structured agentic memory addresses a genuine gap in LLM-based multi-agent planners that typically produce single-shot outputs without systematic revision.
2. The ablation study (Table 5) cleanly isolates each module's contribution, and the constraint transferability experiment (Appendix B.1) provides meaningful evidence that induced rules generalize across structurally different tasks—a non-trivial finding that strengthens the rule library's practical value.
3. Computation overhead is transparently reported (Tables 7–8), and the reduction in planning steps (21.87→18.69) partially amortizes the added per-step cost, demonstrating practical awareness.

Main Concerns
1. The core components are hand-crafted heuristics rather than learned mechanisms, limiting technical novelty. The "evolutionary" generator applies a fixed set of four predefined discrete operators (SwapAgent, InsertAction, DeleteAction, ReplaceObject), and the "symbolic rules" function as shallow correlational templates rather than deep causal representations. There is no mechanism for the system to discover mutation strategies or constraint types beyond these pre-defined bounds—this is an engineering integration of known techniques rather than a fundamental algorithmic contribution.
2. Single-benchmark evaluation with 2–4 agents leaves generalization entirely unsubstantiated. All experiments are on MAP-THOR household rearrangement in single-room layouts. The paper's claim of a "general framework" for multi-agent collaboration is unsupported by any cross-benchmark or cross-domain evidence. The authors' own acknowledgment that the step-budget constraint of MAP-THOR explains the 4-agent performance drop confirms the evaluation's narrow scope rather than mitigating scalability concerns.
3. The GPT-4V (baselines) vs. GPT-4o (EvoCF) discrepancy is an uncontrolled confound. GPT-4o is substantially stronger than the deprecated GPT-4V on vision-language tasks. Without re-running LLaMAR under GPT-4o, the 18% SR improvement cannot be attributed to the framework rather than the model upgrade—this undermines the paper's central quantitative claim.

---

> ### Author Rebuttal · Authors · 2026-03-31
>
> We sincerely thank the reviewer for the constructive feedback and for recognizing the "genuine gap" our work addresses in multi-agent memory-grounded planning. We are heartened by your positive assessment of our systematic design and the transferability of our induced rules, and we have provided additional SAR benchmark results and GPT-4o baseline updates to directly resolve the concerns regarding generalization and model consistency.
>
> ---
> **Weakness 1 & Limitation 3:**
> >*The method relies on **predefined discrete mutation operators** and symbolic rules that act as **shallow templates rather than learned causal abstractions**...*
>
> **Response**:
>
> - **Concern on predefined discrete operators:**
>
> In MAP-THOR, a plan is executed as a sequence of discrete **`Act(agent, object)`** actions. Under this action-level formulation, any alternative plan can differ only in the acting agent, the action/object content, or the presence of an action in the sequence. **Our four operators exactly span these editable degrees of freedom, and are therefore sufficient to express any plan variant, including the optimal one.**
>
> - **Concern on symbolic rules:**
>
> The symbolic constraints are not hand-written rules. EvoCF induces logic-style rules from failure-annotated memory via a **few-shot prompt**; the six relation types are only a lightweight schema.
>
> **`Table 9 (Appendix B.3, Lines 720)`** shows that the induced rules are much richer than the schema itself, covering diverse feasibility and coordination patterns. **`Table 6 (Lines 666–674)`** further shows that rules from a single source task improve unseen target tasks without inducing new rules, indicating reusable structural knowledge beyond shallow heuristics.
>
> ---
> **Weakness 2 & Limitation 1 & Question 3:**
> >*Have you evaluated on any benchmark beyond MAP-THOR?*
>
> **Response**:
>
> We evaluate EvoCF on the **SAR benchmark** from LLaMAR, a partially observable multi-agent search-and-rescue / fire-relief environment that is substantially different from MAP-THOR. As shown in Table 1, across three different SAR scenarios, EvoCF consistently outperforms LLaMAR under the same **GPT-4o** backbone, showing that its gains are **not confined to MAP-THOR**.
>
> |Table 1 (SAR benchmark)|SR|TR|C|B|L|
> |-|-|-|-|-|-|
> |LLaMAR (GPT-4o)|0.75|0.96|0.98|0.82|21.22|
> |EvoCF (GPT-4o)|0.85|0.98|0.99|0.90|19.56|
>
> ---
> **Weakness 3 & Question 1 & Limitation 2:**
> >*The GPT-4V (baselines) vs. GPT-4o (EvoCF) discrepancy is an uncontrolled confound.*
>
> **Response**:
>
> We thank the reviewer for this important point. As shown in table 2, we re-ran LLaMAR with GPT-4o. Under the same **GPT-4o** setup, **LLaMAR improves success rate（SR）from 0.66 to 0.72**, but it still remains below EvoCF’s reported 0.84 SR. Similar gains hold on the auxiliary metrics.
>
> | Table 2|SR| TR|C|B|L|
> |-|-|-|-| -|-|
> |LLaMAR(GPT-4V)|0.66|0.91|0.97|0.82| 21.87|
> |LLaMAR(GPT-4o)|0.72|0.93|0.97|0.85|20.63|
> |EvoCF(GPT-4o)|0.84|0.95|0.99|0.89|18.69|
>
> Our ablation study **`(Table 5, Lines 338-345)`** further shows that the gain comes from **EvoCF itself**, not backbone change alone.
>
> ---
> **Question 2:**
> >*How does the system behave when the optimal strategy requires a coordination pattern outside the four predefined mutation operators...?*
>
> **Response**:
>
> We clarify that, **within EvoCF’s current gene representation, the optimal coordination strategy does not fall outside our operator family**; please see our responses to **`Weakness 1 and Limitation 3`**. This is also reflected empirically in **`Table 3 (line 330)`**: all four mutation operators are used and adopted with non-trivial frequencies.
>
> ---
> **Limitation 4:**
> >*No failure case analysis...*
>
> **Response**:
>
> We provide a **qualitative failure analysis** in **`Section 4.2 (Case Study)`**. That said, we appreciate the reviewer’s suggestion and will add a more **systematic failure-case breakdown / error taxonomy** in a revision.
>
> ---
> **Weakness 2 & Limitation 5:**
> >*The 4-agent performance drop confirms the evaluation's narrow scope...; Scalability beyond 4 agents remains unvalidated*
>
> **Response**:
>
> Thank you for raising this point. **`Table2(Line300)`** is not meant to claim exhaustive scalability, but to test whether EvoCF remains effective as coordination complexity increases.
>
> Since coordination difficulty already rises substantially from 2 to 4 agents, EvoCF’s stable performance without redesign suggests generalization across team sizes. Our claim beyond 4 agents is modest: the current results support robustness under increasing coordination pressure in MAP-THOR, where 5+ agents would be harder instances of the same problem.
>
> Bottlenecks may be less severe without fixed step budgets or strong navigation congestion, though evaluating such settings is beyond the current benchmark.
>
> ---
> We hope these clarifications on our mutation operators and the inclusion of cross-domain results provide a more comprehensive view of EvoCF’s scalability, and we look forward to the discussion phase.

---

> > ### Author Rebuttal · Reviewer_hm4U · 2026-04-03
> >
> > Thank you for the authors' response. I will raise my score.

---

> > > ### Author Response · Authors · 2026-04-04
> > >
> > > We sincerely appreciate the thoughtful and concrete suggestions you raised. Your comments have prompted us to examine several important aspects of the method more carefully, and we believe they have meaningfully strengthened the revised manuscript.  We're glad the revisions addressed your concerns, and appreciate your support in updating the score. Wishing you all the best！

---

### Official Review · Reviewer_EWpR · 2026-03-13

**Soundness:** 2
**Presentation:** 1
**Significance:** 2
**Originality:** 2
**Overall Recommendation:** 3
**Confidence:** 4

**Summary:**

The paper proposes EvoCF, an agentic-memory-driven evolutionary counterfactual planning framework for multi-agent embodied collaboration. The method first induces symbolic constraints from failure-annotated execution traces, then systematically explores alternative joint plans by mutating action assignments and ordering, and finally uses an agentic memory-grounded evaluator to rank candidate plans using both learned constraints and retrieved outcomes. Experiments on MAP-THOR show the effectiveness of the proposed method.

**Compliance With Llm Reviewing Policy:**

Affirmed.

**Final Justification:**

I appreciate the author's efforts in rebuttal. While part of my initial concerns about the presentation are addressed during the rebuttal, my concerns over the weak evaluation remain (out-of-date llm backbones, single benchmark with a simulator, additional evaluation on a grid env only), and a major rewriting of the experiments section would be needed. Therefore, I maintain my score of weak reject.

**Key Questions For Authors:**

- The Agentic-Memory-Grounded Evaluation is confusing to me; Why is it called agentic? Is there a tool to call or what? Is it an agentic module in an embodied agent?
- The symbolic constraints seem manually defined, which may constrain the method's applicability.
- Experiments are conducted on a single benchmark, while other multi-agent embodied collaboration benchmark exists.

**Limitations:**

yes

**Strengths And Weaknesses:**

Strengths:
- Introducing counterfactual re-planning into multi-agent collaboration is interesting.
- The ablation studies and experiments across different numbers of agents support the central claim well.

Weaknesses:
- The method is hard to understand
  - l173: It's confusing to represent so many different things in one symbol \psi, which makes the later method section also hard to follow.
  - The Agentic-Memory-Grounded Evaluation is confusing to me; Why is it called agentic? Is there a tool to call or what? Is it an agentic module in an embodied agent? Sounds weird.
- The symbolic constraints seem manually defined, which may constrain the method's applicability.
- Experiments are conducted on a single benchmark, while other multi-agent embodied collaboration benchmark exists, e.g.[1].
- The results section (4.1) could be better organized; it's very confusing now.
- Minor:
  - Typo in Figure 1: Memory-Griven Evaluator
  - Still ? in l290, l303
  - Table 3 captioning is hard to understand

[1] Building cooperative embodied agents modularly with large language models. ICLR24

---

> ### Author Rebuttal · Authors · 2026-03-31
>
> We appreciate the Reviewer’s detailed assessment and the opportunity to improve the clarity of our framework. To address the concerns regarding presentation and terminology, we have refined our notation, clarified the "agentic memory" formulation, and included additional cross-domain experiments that demonstrate EvoCF’s effectiveness beyond the MAP-THOR environment.
>
> ---
> **Weakness 1:**
> >*It's confusing to represent so many different things in one symbol \psi...*
>
> **Response**:
>
> The following clarifies the distinctions among key symbols and concepts:
>
> ```
> [Memory 𝓜]
> └── stores episodic transitions m = (oₜ, aₜ, {ψₖ})
>     └── ψₖ ∈ Ψ (line 118): annotation type space
>         → e.g., ψ_plan, ψ_eff, ψ_pre, ψ_fail (line 120)
> ```
> - **Memory (𝓜)** is the structured episodic store of transition-level records and annotations;
> - **ψ (line 122)** refers to structured annotations in memory (e.g., `ψ_plan`, `ψ_fail`,  `ψ_pre,`,`ψ_eff`) that encode subgoal IDs, observed effects, or failure signals from past episodes.
>
> ---
> **Weakness 2 & Question 1:**
> >*The Agentic-Memory-Grounded Evaluation is confusing to me*
>
> **Response**:
>
> In our setting, **agentic memory refers to memory being actively integrated into the decision loop, rather than a passive log or a separate agent.** This usage is consistent with recent survey work[1], which defines agentic memory as memory that directly participates in decision-making.
>
> **Concretely, EvoCF abstracts past executions into reusable symbolic constraints for planning-time counterfactual generation. At evaluation, it retrieves relevant transitions and constraints, instead of using forward simulation, and lets the LLM judge feasibility and coordination quality.**
>
> This design makes the evaluator effectively **world-model-free**, as it relies on experience-grounded evidence and abstracted rules, rather than explicit dynamics modeling.
>
> ---
> **Weakness 3 & Question 2:**
> >*The symbolic constraints seem manually defined, which may constrain the method's applicability.*
>
> **Response**:
>
> The rule library in EvoCF is automatically induced, not hand-written. The six relation types are only a lightweight induction schema, while the actual logic-style constraints are generated by a **few-shot prompt** from failure-annotated memory. As shown in **`Table 9(Appendix B.3, Line 720)`**, this yields a much richer and more diverse constraint library than the schema itself.
>
> ---
> **Weakness 4 & Question 3:**
> >*Experiments are conducted on a single benchmark, while other multi-agent embodied collaboration benchmark exists, e.g.CoELA[2].*
>
> **Response**:
> Thank you for the helpful reference. However, CoELA is not directly comparable to EvoCF, because the two methods are evaluated under different system scopes and task settings
>
> We evaluate EvoCF on the **SAR benchmark** from LLaMAR, a partially observable multi-agent search-and-rescue / fire-relief environment that is substantially different from MAP-THOR. As shown in Table 1, EvoCF consistently outperforms LLaMAR under the same **GPT-4o** backbone, showing that its gains are **not confined to MAP-THOR**.
>
> |Table 1 (SAR benchmark)|SR|TR|C|B|L|
> |-|-|-|-|-|-|
> |LLaMAR (GPT-4o)|0.75|0.96|0.98|0.82|21.22|
> |EvoCF (GPT-4o)|0.85|0.98|0.99|0.90|19.56|
>
> ---
> **Weakness 5:**
> >*The results section (4.1) could be better organized.*
>
> **Response**:
>
> Thank you for this helpful suggestion. In the revision, we will reorganize Sec. 4.1 into a clearer question-driven structure with explicit subheadings: (Q1) overall comparison against baselines, (Q2) scalability with different numbers of agents, (Q3) component-level analysis explaining where the gains come from, and (Q4) sensitivity to retrieval depth. Concretely, we will reorder the discussion so that Table 1 addresses overall effectiveness, Table 2 addresses scaling, Table 5 and Table 3 explain the contribution of the main modules and mutation behavior, and Table 4 presents sensitivity analysis. We will also add short take-home statements at the start of each subsection so that each table answers one clear question.
>
> ---
> **Minor 1&2:**
> >*Typos*
>
> **Response**:
>
> Thank you for pointing out these. We will correct the typo in the revision.
>
> ---
> **Minor 3:**
> >*Table 3 captioning is hard to understand*
>
> **Response**:
>
> We will revise the caption for Table 3 to ensure the frequency metrics are explicitly defined and their relationship is clear.
>
> Specifically, **Occurrence Frequency** denotes how often each operator appears in generated counterfactual candidates, while **Adoption Frequency** represents how often a counterfactual candidate generated by that operator is finally selected for execution.
>
> ---
> [1] Memory in the Age of AI Agents: A Survey. 2026
>
> [2] Building cooperative embodied agents modularly with large language models. ICLR24
>
> ---
> We believe these clarifications provide a more intuitive grounding of our notation, and we remain available for further technical discussion to assist in your re-evaluation.

---

> > ### Author Rebuttal · Reviewer_EWpR · 2026-04-02
> >
> > Thank you for the clarifications and additional experiments. My concerns over presentation are partially resolved now. However, after reading other reviews, I think one major concern is the evaluation. The results from the main table (Table 1), except for the proposed method, seem directly taken from the llamar paper, without any notations, and without any efforts to reproduce them with the same modern llm backbones for fairer comparison. This raises serious concerns over the experiments. The newly introduced SAR Benchmark is also adopted from the llamar paper, without any details. I have to go read llamar paper for understanding what this benchmark is, which seems to be a grid world-based environment, significantly easier than MAP-THOR, or benchmarks in CoELA I suggested. Overall, I think the idea is interesting, but a major revision to the experiments and the presentation would be needed before re-evaluation could be made, therefore I keep my negative recommendation.

---

> > > ### Author Response · Authors · 2026-04-03
> > >
> > > We thank the reviewer for the thoughtful follow-up comments. We understand the reviewer’s additional concerns regarding the fairness and scope of the evaluation, and we provide further clarification on these points below.
> > >
> > > ---
> > > >***Concern on fairer comparison without the same modern llm backbones.***
> > >
> > > **Response**:
> > >
> > > We thank the reviewer for raising this important concern. We agree that comparing methods under mismatched backbone settings would weaken the fairness of the evaluation. **In response to this concern raised by both you and other reviewers, we have re-ran LLaMAR with the same GPT-4o backbone used by EvoCF**.
> > >
> > > | Table 1|SR| TR|C|B|L$\downarrow$|
> > > |-|-|-|-| -|-|
> > > |LLaMAR(original)|0.66|0.91|0.97|0.82| 21.87|
> > > |LLaMAR(GPT-4o)|0.72|0.93|0.97|0.85|20.63|
> > > |EvoCF(GPT-4o)|0.84|0.95|0.99|0.89|18.69|
> > >
> > > As shown in Table 1, Under the same **GPT-4o** setup, LLaMAR improves success rate（SR）from 0.66 to 0.72, **but it still remains below EvoCF’s reported 0.84 SR**. Similar gains hold on the auxiliary metrics.
> > >
> > > To further verify that the benefit is **method-driven rather than tied to a single modern LLM**, we have additionally evaluated both **LLaMAR and EvoCF** on two other backbone-matched settings: **Claude 3.5 Sonnet** and **GPT-4o mini**.
> > >
> > > |Table 2|SR|TR|C|B|L$\downarrow$|
> > > |-|-|-|-|-|-|
> > > |LLaMAR (Claude 3.5 Sonnet)|0.70|0.92|0.96|0.84|21.12|
> > > |EvoCF (Claude 3.5 Sonnet)|0.81|0.94|0.98|0.87|19.18|
> > > |LLaMAR (GPT-4o mini)|0.61|0.90|0.96|0.81|22.34|
> > > |EvoCF (GPT-4o mini)|0.74|0.93|0.98|0.85|20.11|
> > >
> > > As shown in Table 2, **EvoCF consistently outperforms the corresponding LLaMAR baseline on both backbones.** Notably, even with the weaker **GPT-4o mini** backbone, EvoCF achieves performance that is **on par with or slightly better than LLaMAR under GPT-4o**. Together, these results show that EvoCF’s advantage is robust across multiple modern backbones, rather than an artifact of evaluating our method with a stronger model.
> > >
> > > >***Concern on SAR benchmark.***
> > >
> > > **Response**:
> > >
> > > Due to the rebuttal character limit, our earlier response did not have space to provide a fuller description of the SAR benchmark.
> > >
> > > In the original LLaMAR paper, SAR is a **partially observable multi-agent search-and-rescue / fire-relief environment** where agents must find missing people in unknown locations, extinguish spreading **Class A/B fires** using the correct resources (**water/sand**) gathered from distributed reservoirs, and **coordinate to carry rescued people to a drop-off point**, with each rescue requiring **two more agents simultaneously**.
> > >
> > > By contrast, SAR is not another household variant, but a different domain with different state/action semantics and a different source of difficulty: **dynamic hazards, resource dependencies, hidden targets, and explicit multi-agent coordination constraints**. In that sense, we used SAR not as a cosmetic extra benchmark, but as a deliberately different testbed for cross-domain generalization.
> > >
> > > **Although SAR is grid-based**, we believe it still provides a meaningful and non-trivial test of generalization rather than a strictly easier setting. Its **visual representation is simpler**, but its **high-level planning problem is in several respects harder**.
> > > - First, SAR is **time-critical and non-stationary**: fires spread over time, and the spread rate increases with intensity, so delays or suboptimal action ordering can create **irreversible future consequences**.
> > > - Second, tasks require longer **causal dependency chains**: agents must identify the fire source and type, obtain the correct resource, navigate to the right location, and deploy it in time.
> > > - Third, SAR imposes **stronger explicit coordination** than MAP-THOR in key cases, since rescuing a person requires synchronous multi-agent carrying and drop-off.
> > > - Fourth, agents face a genuine **task-allocation trade-off** between exploring for hidden people and suppressing active fires before they become uncontrollable. These are not superficial complications; they directly increase the sequential decision-making burden.
> > >
> > > Indeed, the LLaMAR paper explicitly motivates SAR in terms of **explicit cooperation**, **exploration-vs-intervention trade-offs**, **uncertainty**, and **irreversibility**, and its reported SAR failure modes include **incorrect causal ordering** and **too-late action sequences** under fire spread. So while SAR is less visually rich than MAP-THOR, it is arguably **more challenging in coordination and long-horizon decision-making**. For our purposes, this is precisely why it is a meaningful benchmark: success on SAR shows that EvoCF is not limited to household rearrangement priors, but transfers to a qualitatively different partially observable domain where coordination mistakes are often even more consequential.
> > >
> > > ---
> > > We hope these additional clarifications and results help resolve the reviewer’s concerns. We would be grateful for the reviewer’s further consideration and any updated assessment in light of these additions.

---

### Official Review · Reviewer_CbGe · 2026-03-17

**Soundness:** 2
**Presentation:** 2
**Significance:** 2
**Originality:** 3
**Overall Recommendation:** 4
**Confidence:** 3

**Summary:**

This paper proposes EvoCF, a framework for embodied collaboration that uses counterfactual plan generation guided by symbolic constraints extracted from past failures. EvoCF introduces three modules: (1) a Symbolic Constraint Inductor that extracts reusable rules from failure experiences and accumulates them into an evolving rule library; (2) an Evolutionary Counterfactual Plan Generator that produces alternative joint plans by applying four mutation operators to an initial plan, guided by retrieved constraints; and (3) an Agentic Memory-Grounded Evaluator that ranks candidate plans by retrieving similar past transitions and reasoning over constraint satisfaction and historical outcomes.EvoCF achieves 84% success rate on the MAP-THOR benchmark.

**Compliance With Llm Reviewing Policy:**

Affirmed.

**Final Justification:**

The rebuttal provides additional experiments that address most of my concerns and demonstrate solid empirical gains. That said, several design choices still seem more empirical than principled, and the work could benefit from further polishing in this regard.

**Key Questions For Authors:**

Check weakness above.

**Limitations:**

The authors had discussed that the base model is different which is great.  Several methodological limitations deserve explicit discussion. First, the experimental scale is limited: all evaluation is conducted on a single benchmark (MAP-THOR) with 45 tasks in single-room layouts, and no results are reported on other environments or task domains. Second, the system's dependence on RAG-based retrieval introduces a granularity mismatch: memory records are stored at the single-timestep transition level, but many coordination failures are caused by decisions made several steps earlier. When a failure manifests at step 8 due to a poor assignment at step 5, the retrieval finds records matching step 8's surface features rather than the root cause, limiting the system's ability to learn from structurally complex failures. Third, the counterfactual generator lacks any explicit diversity mechanism. The LLM is prompted to produce N variants, but nothing prevents it from generating N very similar candidates. Without novelty pressure or anti-redundancy filtering, the candidate set may cluster in a narrow region of the plan space, reducing the value of the counterfactual search.

**Strengths And Weaknesses:**

Strengths:
- [s1] The core insight that LLM planners produce single-shot plans without systematic revision is valid, and the idea of explicitly exploring counterfactual alternatives is well-motivated for multi-agent settings where coordination failures are common.
- [s2] Constraint induction from failures is a valuable and transferable idea. The constraint transfer experiment (Table 6) is the paper's strongest evidence: rules induced from "put food in the fridge" improve success rates on "put pots on the stove" (0.60→0.80) and "open all drawers" (0.40→0.60), without any task-specific counterfactual generation. This demonstrates that the rules capture genuine coordination knowledge
- [s3] Clear ablation in sec 4.2 to showcase the impact of each component.

Weakness:
- [w1] All baselines use GPT-4V (deprecated), while EvoCF uses GPT-4o. The LLM backbone is the single most important variable in LLM-based planning systems. The 18-point improvement over LLaMAR could partly or largely reflect the model upgrade rather than the EvoCF methodology. At minimum, LLaMAR must be re-run with GPT-4o. Without this, the absolute magnitude of improvement is unreliable.

- [w2] "Evolutionary" framing is a bit of an overclaim. There is no population maintenance, no crossover, no multi-generation iteration, no fitness-proportional selection. This is a single-round constrained generate-and-rank. The "gene representation" relabels action assignments without adding modeling capacity. Calling an action a "gene" does not introduce genotype-phenotype distinction, linkage, or any evolutionary structure. The paper should either implement genuine multi-generation search or adopt accurate terminology.

- [w3] Several references use “?” instead of proper citations (lines 288, 290, 299).

- [w4] Rules are added via deduplication only. Semantically equivalent but differently worded rules could accumulate as duplicates.

- [w5] The pipeline is tested on only one backbone agent. It would be beneficial to show EvoCF is a general framework across different models.

---

> ### Author Rebuttal · Authors · 2026-03-31
>
> We appreciate the reviewer’s thoughtful and detailed feedback. In this response, we provide comprehensive evaluations on GPT-4o and the SAR benchmark, along with new cross-model results, to directly address the concerns regarding model consistency and framework generalization.
>
> ---
> **Weakness 1:**
> >*LLaMAR must be re-run with GPT-4o.*
>
> **Response**:
>
> We thank the reviewer for this important point. As shown in table 1, we re-ran LLaMAR with GPT-4o. Under the same **GPT-4o** setup, **LLaMAR improves success rate（SR）from 0.66 to 0.72**, but it still remains below EvoCF’s reported 0.84 SR. Similar gains hold on the auxiliary metrics.
>
> | Table 1|SR| TR|C|B|L$\downarrow$|
> |-|-|-|-| -|-|
> |LLaMAR(GPT-4V)|0.66|0.91|0.97|0.82| 21.87|
> |LLaMAR(GPT-4o)|0.72|0.93|0.97|0.85|20.63|
> |EvoCF(GPT-4o)|0.84|0.95|0.99|0.89|18.69|
>
> **`Table 5 (Lines 338)`** further shows that the gain stems from **EvoCF itself**, rather than backbone changes.
>
> ---
> **Weakness 2:**
> >*Concern on evolutionary framing.*
>
> **Response**:
>
> We agree that our original terminology may overstate the connection to classical evolutionary algorithms.
>
> From this perspective, our use of **“evolutionary”** aligns with emerging paradigms of **LLM-based optimization** over **structured artifacts** (e.g., GEPA[1], ICLR 2026 Oral). EvoCF instantiates this principle for **online multi-agent plan refinement under an evolving rule library**.
>
> In our setting, maintaining a persistent **population** or **multi-generation optimization** is often impractical, as plan quality depends strictly on the **current observation** and **coordination context**. Consequently, EvoCF focuses on **localized, rule-conditioned refinement of executable plans**.
>
> We will revise the terminology (e.g., “evolutionary-inspired counterfactual refinement”) in the final version.
>
> [1]GEPA: Reflective Prompt Evolution Can Outperform Reinforcement Learning
>
> ---
> **Weakness 3:**
> >*Several references use “?”.*
>
> **Response**:
>
> Thank you for catching this. We will correct these citations in the revision.
>
> ---
> **Weakness 4:**
> >*Rules are added via deduplication only.*
>
> **Response**:
>
> We agree that deduplication alone does not remove all semantically equivalent rules. In EvoCF, however, **rules are retrieved and applied softly in embedding space, rather than used as rigid discrete entries, so similar rules are usually handled consistently at inference time.**
>
> ---
> **Weakness 5:**
> >*The pipeline is tested on only one backbone agent.*
>
> **Response**:
>
> We evaluate EvoCF on **10 tasks** with **Claude 3.5 Sonnet** and **GPT-4o mini**. As shown in Table 2, EvoCF consistently outperforms the backbone-matched LLaMAR baseline on both backbones, suggesting that EvoCF is **not tied to a single backbone**.
>
> |Table 2|SR|TR|C|B|L$\downarrow$|
> |-|-|-|-|-|-|
> |LLaMAR (Claude 3.5 Sonnet)|0.70|0.92|0.96|0.84|21.12|
> |EvoCF (Claude 3.5 Sonnet)|0.81|0.94|0.98|0.87|19.18|
> |LLaMAR (GPT-4o mini)|0.61|0.90|0.96|0.81|22.34|
> |EvoCF (GPT-4o mini)|0.74|0.93|0.98|0.85|20.11|
>
> ---
> **Limitations 1:**
> >*No results are reported on other environments.*
>
> **Response**:
>
> We evaluate EvoCF on the **SAR benchmark** from LLaMAR, a partially observable multi-agent search-and-rescue / fire-relief environment substantially different from MAP-THOR. As shown in Table 3, across three different SAR scenarios, EvoCF consistently outperforms LLaMAR under the same **GPT-4o** backbone, showing that its gains are **not confined to MAP-THOR**.
>
> |Table 3 (SAR benchmark)|SR|TR|C|B|L$\downarrow$|
> |-|-|-|-|-|-|
> |LLaMAR (GPT-4o)|0.75|0.96|0.98|0.82|21.22|
> |EvoCF (GPT-4o)|0.85|0.98|0.99|0.90|19.56|
>
> ---
> **Limitations 2:**
> >*Concern on **retrieval granularity**.*
>
> **Response**:
>
> We agree that transition-level storage may appear insufficient for long-horizon failures. However, EvoCF does not rely on raw transitions alone. During constraint induction, the LLM reasons over the current transition together with multi-step context, including task state, agent roles, and preceding observations in the POMDP trajectory. As a result, the induced symbolic constraints often capture coordination dependencies or violated preconditions, rather than only the surface symptom at the failure step.
>
> Thus, transition-level memory serves as an anchor, while the learned constraints encode higher-level structure that can generalize across steps.
>
> ---
> **Limitations 3:**
> >*"Concern on **candidate plan diversity**.*
>
> **Response**:
> The counterfactual plan generator is not asked to freely paraphrase plans. **`The generator prompt（Appendix A, Lines 597-598）`** requires each candidate to be constructed by applying **one or more mutation operators to one or more agents**, which already pushes candidates toward different **failure-repair modes** rather than near-duplicate rewrites.
>
> ---
> We hope these additional results and structural clarifications address your concerns, and we welcome any further discussion to help us refine the final version of this work.

---

> > ### Author Rebuttal · Reviewer_CbGe · 2026-04-03
> >
> > Thanks to the author for the rebuttal and additional experiments. I would raise the score to 4.
> > That said, I still have a remaining concern regarding the deduplicate operation, and over many episodes, this could degrade retrieval quality by consuming top-k slots with redundant constraints rather than surfacing diverse ones. It would be helpful if the authors could report how the rule library size grows over episodes and whether retrieval precision is affected as the library scales. A more principled consolidation mechanism would strengthen the framework's long-term scalability.

---

> > > ### Author Response · Authors · 2026-04-04
> > >
> > > Thank you for your thoughtful review and encouraging response. In response to your concerns, we provide our specific clarifications below.
> > >
> > > ---
> > > **Concern on rule-library growth over episodes.**
> > > The rule library grows online by adding only newly induced constraints from failure-annotated transitions through ($R \leftarrow \mathrm{Dedup}(R \cup C_{\text{gen}}(m))$), rather than appending all generated rules indiscriminately. As a result, its size does not grow linearly with the number of episodes, since recurring failure patterns are mapped back to existing rule forms under deduplication.
> > >
> > > **Concern on retrieval quality as the library scales.**
> > > In EvoCF, retrieval is performed over **top-k memory entries**, and the attached symbolic rules are aggregated only after those experiences are retrieved, so a larger rule library does not directly imply that redundant rules will individually consume retrieval slots. Empirically, our retrieval-depth sensitivity analysis shows that performance improves from **(K=1)** to **(K=5)** and only drops slightly at **(K=10)**, which suggests that redundancy or noise may begin to appear at larger retrieval sets but is not yet a dominant issue at the current scale.
> > >
> > > In the revision, we will provide further analysis and discussion of these issues. And we fully agree that a more principled consolidation mechanism would strengthen the framework’s long-term scalability, we view this as an important direction for further exploration.
> > >
> > > ---
> > > We truly appreciate your recognition of our additional efforts. If you have any further suggestions, it would be our pleasure to hear them. Best wishes for your future work and life.

---

### Decision · Program_Chairs · 2026-04-30

**Decision:**

Accept (regular)

**Comment:**

This paper proposes EvoCF, a framework for multi-agent embodied collaboration that induces symbolic constraints from past failures and uses them to guide counterfactual joint-plan refinement through structured mutation operators. The motivation is clear in that existing LLM-based planners tend to produce one-shot plans that poorly handle coordination constraints and physical dependencies, and the paper targets this gap directly.

Reviewers broadly found the core idea well-motivated, with particular appreciation for the constraint transferability experiment and the clean ablation study.

The reviews raised concerns in several areas such as backbone mismatch between baselines and the proposed method, limited benchmark coverage, the overreach of the evolutionary framing, and questions about rule library scalability over time. The rebuttal addressed most of these. The authors ran backbone-matched baselines and added evaluation on the SAR benchmark. Accordingly, two reviewers moved from 3 to 4 as a result, and one was consistently positive throughout.

Yet, one reviewer maintained Weak Reject, considering that the evaluation is still confined to simulation and that grid-based SAR does not make a compelling case for broader generalization. Another reviewer would like to see a more principled treatment of rule library growth and its downstream effect on retrieval quality.

With a final score distribution of 3/4/4/4 and a positive shift after rebuttal, I recommend Weak Accept.